# Review of Patient Gene Profiles Obtained through a Non-Negative Matrix Factorization-Based Framework to Determine the Role Inflammation Plays in Neuroblastoma Pathogenesis

**DOI:** 10.3390/ijms25084406

**Published:** 2024-04-17

**Authors:** Angelina Boccarelli, Nicoletta Del Buono, Flavia Esposito

**Affiliations:** 1Department of Precision and Regenerative Medicine and Polo Jonico, School of Medicine, University of Bari Aldo Moro, Piazza Giulio Cesare 11, 70121 Bari, Italy; angelina.boccarelli@uniba.it; 2Department of Mathematics, University of Bari Aldo Moro, Via Edoardo Orabona 4, 70125 Bari, Italy; flavia.esposito@uniba.it

**Keywords:** neuroblastoma, inflammatory phenotype, gene profiling, biomarkers

## Abstract

Neuroblastoma is the most common extracranial solid tumor in children. It is a highly heterogeneous tumor consisting of different subcellular types and genetic abnormalities. Literature data confirm the biological and clinical complexity of this cancer, which requires a wider availability of gene targets for the implementation of personalized therapy. This paper presents a study of neuroblastoma samples from primary tumors of untreated patients. The focus of this analysis is to evaluate the impact that the inflammatory process may have on the pathogenesis of neuroblastoma. Eighty-eight gene profiles were selected and analyzed using a non-negative matrix factorization framework to extract a subset of genes relevant to the identification of an inflammatory phenotype, whose targets (*PIK3CG*, *NFATC2*, *PIK3R2*, *VAV1*, *RAC2*, *COL6A2*, *COL6A3*, *COL12A1*, *COL14A1*, *ITGAL*, *ITGB7*, *FOS*, *PTGS2*, *PTPRC*, *ITPR3*) allow further investigation. Based on the genetic signals automatically derived from the data used, neuroblastoma could be classified according to stage rather than as a “cold” or “poorly immunogenic” tumor.

## 1. Introduction

Neuroblastoma (NB) is an extracranial solid tumor, and it is considered the most frequently diagnosed neoplasia in children (7–10%) after leukemia and brain tumors [1,2]. It originates from neural crest cells (NCCs), multipotent cells close to the neural tube, which migrate and differentiate into multiple cell types during embryogenesis, including sympathetic neurons and adrenal medullary cells (modified postganglionic sympathetic neurons). Deregulation of signaling pathways involved in NCC cell differentiation can lead to the development of NB anywhere in the sympathetic nervous system or the medullary region of the adrenal gland. NB is a highly heterogeneous tumor in terms of clinical behavior, consisting of different sub-cell types and showing a variety of genomic alterations such as MYCN amplification, different DNA ploidy patterns, deletion of the short arm of chromosome 1, gain of arm of chromosome 17q, and deletion of chromosome 11q. These genomic alterations, histopathology, age, and stage according to the International Neuroblastoma Staging System (INSS) are used to classify patients with NB into risk groups, including low, intermediate, and high risk [3]. Patients with high-risk NB are challenging to treat and require more effective therapeutic approaches that require a deeper understanding of the complex intratumoral heterogeneity of NB [4]. As few genetic defects have been identified, most recent research has focused on identifying the molecular pathways involved in the complex pathogenesis of NB. Several signaling pathways have been identified in the literature that contribute to the growth and progression of NB and its resistance to conventional treatments [5]. It is possible to schematize the role of signaling pathways involved in the pathophysiology of NB as follows:(i)MYCN signaling promoting NB cell proliferation and activating MDM2 expression.(ii)Wnt signaling, which is involved in stemness and increases the level of MYCN.(iii)ALK signaling, which activates the expression of PI3K/AKT/mTOR, RAS-MAPK, and MYCN.(iv)p53-MDM2 pathway, which promotes angiogenesis, MYCN translation and drug resistance.(v)PI3K/AKT/mTOR pathway which promotes survival and chemoresistance of NB cells.(vi)RAS-MAPK signaling, which is activated by EGFR, promotes neuroblastoma cell survival.(vii)TrkB signaling activating PI3K/mTOR [6,7].

Rapid advances in omics techniques and “big data” analysis have significantly increased knowledge of the molecular landscape of NB, and its molecular profiling has provided bioinformatics pipelines to guide different therapeutic options. The recent discovery that two distinct epigenetic states, the less differentiated mesenchymal and neural crest-like (MES) and the more differentiated adrenergic or sympathetic noradrenergic (ADRN), coexist and can spontaneously switch between each other represents an advance in the understanding of NB biology [8,9,10]. In primary human neuroblastoma, ADRN status identifies most of the tumor at diagnosis [8,9], whereas MES status is enriched at the time of recurrence and metastatic disease [11].

Further data provided by the literature define NB as a “cold tumor” because it lacks basal inflammatory signaling, T-cell infiltration, and a very low expression of the major histocompatibility complex class I (MHC-I) [12]. This information indicates how the genetic complexity of NB requires further investigations; in fact, further attempts should be made to select new signaling pathways that may be responsible for the pathogenesis. The definition of NB as a “cold” tumor has aroused our curiosity enough to want to understand this condition; in particular, the aim of our study is to identify the protagonists of inflammatory events and the potential pathways in which they are involved, starting with NB samples derived from primary tumors of untreated patients whose gene profiles are available. If the answer to our question is affirmative, we would like to identify additional prognostic tumor markers that could be helpful in determining a patient’s prognosis and/or response to treatment, in addition to those that are currently known from the literature. Since children make up many NB patients, finding novel tumor markers is crucial to using them as therapeutic targets and creating customized therapy based on a patient’s genetic predisposition.

The paper is organized as follows: Section 2 details the pathway enrichment analysis of extracted genes, discusses the results for understanding potential biomarkers associated with NB cancer, and provides some gene association network analysis (performed using GeneMANIA and STRING software) that further consolidated the validity of these extracted genes, Section 3 describes the NB samples included in this study and the extraction framework used to identify the subset of genes extracted and discussed in Section 2.

## 2. Results and Discussions

The subset of genes extracted from Metagene4 (as described in Section 3) was analyzed by pathway enrichment analysis with the WebGeastalt tool [13], using the reference database PANTHER (Analysis of proteins through evolutionary relationships, http://www.pantherdb.org, accessed on 30 November 2022), which combines genomes, gene function classifications, pathways, and statistical analysis tools to cluster genes into functional pathways that also outline the molecular functions and biological processes under investigation. These genes are analyzed according to the following parameters: the reference organism: hsapiens; enrichment categories: Panther pathways; type ID: gene symbol; reference list: genome-protein-coding. The enrichment analysis was performed with a minimum and maximum number of IDs. The statistical method chosen for the FDR was BH (Yoav Benjamini and Yosef Hochberg BH procedure).

Table 1 reports the ten pathways selected; the first six pathways have a *p*-value ≤ 0.05 and an FDR ≤ 1 and contain 79 genes (representing 10.2% of the 771 extracted genes from Metagene4) of which 15 are shared in the different pathways; the remaining enrich the single pathways.

The analysis of the 88 primary NB tumor profiles allows the selection of the 79 genes listed in Table 2. These genes are distributed in 6 of the 10 pathways reported in Table 1 and meet the significance criteria FDR < 1 and *p*-value < 0.05.

Figure 1 reports the circularly composited visualizations of genes and pathways and the indications of the shared genes in each of the six selected pathways.

These data suggest that although NB is a “cold” and/or “non-immunogenic tumor” [12], the inflammatory process—mediated by the chemokine and cytokine signaling pathway—undoubtedly plays a critical role in the progression of the disease. In tumors, inflammatory pathways are essential for activating signaling and establishing a permissive microenvironment. A pediatric NB neoplasm is characterized by functional abnormalities associated with two genes: the *PIK3CG* gene shared by five pathways (T-cell activation, inflammation mediated by chemokine and cytokine signaling pathway, integrin signaling pathway, B-cell activation, axon guidance mediated by netrin) and the *NFATC2* gene, shared by four pathways (T-cell activation, inflammation mediated by chemokine and cytokine signaling pathway, B-cell activation, axon guidance mediated by netrin).

The *PIK3CG* gene encodes for a class I catalytic subunit of *PI3K* and plays the role of a “hub”; i.e., it collects numerous extracellular signals to send them downstream through the regulatory action of the other components *PIK3R2* (T-cell activation, integrin signaling pathway, axon guidance mediated by netrin) and *PIK3R5* (netrin-mediated axon guidance) This gene also induces pleiotropic actions. In fact, the signals transmitted by the *PIK3CG* protein could be necessary for the recruitment of inflammatory cells in the tumor microenvironment through the adhesion mediated by integrins (*ITGB7*, *ITGAL*) and chemokine receptors (*CCR1*, *CCR2*, *CCR5*, *CCR7*, *CX3CR1)*, in response to growth factors and to the same chemokines (*CCL11*, *CCL13*, *CCL18*, *CCL2*, *CCL21*, *CCL4*, *CCL5*, *CCL8*, *CXCL10)*. *PIK3CG* signaling can effectively promote angiogenesis and tumor-associated inflammation [15]; could affect the *NFATC2* gene, which encodes a calcium-sensitive protein; and is characterized by its role in immune responses [16] and neural development [17]. *NFATC2* was first identified as a transcriptional regulator in T cells, but it also has a major impact on tumor growth, metastasis, and the creation of microenvironments that enable tumor growth [16,18]. The T-cell activation pathway selected by pathway enrichment analysis revealed that the role of *NFATC2* in the immune response is not limited to transcriptional regulation of T cells but also involves the following:i.B-cell activation.ii.Axon growth guidance mediated by netrins, which act as growth factors and promote growth activity in target cells (netrin-mediated axon guidance) [19].

The activation of *NFATC2* could be initiated by the integrins *ITGB7* and *ITGAL*, stimulated by extracellular matrix (ECM) elements such as collagenases (*COL6A2*, *COL6A3*, *COL12A1*, *COL14A1*), and the receptors *FPR1* and *FPR3*, which, among other activation pathways, induce Ca2+ release in NB cells via *ITPR3* (inflammation mediated by chemokines and cytokines signaling pathway) and signal transducers. These stimuli involve the translocation of NFATC2 to the nucleus, which is activated by the formation of the AP-1 complex, which includes both *JUNB* (chemokine and cytokine signaling pathway) and *FOS*. In T cells, AP-1 forms ternary complexes with *NFATC2* and binds *NFATC2/AP-1* sites present in regulatory regions of cytokine and effector genes [20].

The proliferation, migration, and angiogenesis of tumor cells are highly dependent on the *NFATC2* protein [16], which can also directly activate the transcriptional activity of the oncogene by binding to the proximal promoter of the *MYC* oncogene. *NFATC2* silencing or pharmacological inhibition downregulates *MYC* and some of its targets. To facilitate epithelial-to-mesenchymal transition (*EMT*), it has been hypothesized that *NFATC2* is also functionally related to genes that regulate mesenchymal programs in cancer cells [21].

*NFATC2* transcribes *PTGS2*, which is involved in inflammatory processes; the *PTGS2* protein is secreted into the extracellular matrix, catalyzing the production of the pro-inflammatory prostaglandin *PGE2* from arachidonic acid (AA). The latter is also metabolized by lipoxygenases (ALOX5AP) to produce chemoattractants (*CCL2*, *CCL4*, *CXCL10*) [22]. *PGE2* has both autocrine and paracrine effects and is a strong inducer of the migration and proliferation of tumor cells. *PGE2* diffuses throughout the brain parenchyma and activates the immune system, endothelial cell proliferation, vessel formation, and migration differently depending on which receptors it interacts with [16,23]. In the tumor microenvironment, the increased production of *PGE2* (caused by *PTGS2* activation) determines the increased expression of indoleamine 2,3-dioxygenase (IDO1) which activates the kynurenine pathway where tryptophan is used to produce serotonin and melatonin promoting oxidative stress [24] (see Appendix A). The toll-like receptors *TLR2*, *TLR3*, and *TLR7*, along with *CD14* and *LY96*, enhance the toll receptor signaling pathway when the *PTGS2* gene is expressed in NB cells.

The two genes *LY96* and *CD14* are activated and act in concert in response to lipopolysaccharide (LPS) stimuli with the cell surface toll-like receptor 4 (TLR4), providing a link between the receptor and LPS signaling. The transcription (*NFATC2)* of pro-inflammatory cytokines (*PTGS2*, *PGE2*) and chemokines with chemotactic (*CCL5*, *CCL4*, *CXCL10*) and co-stimulatory (*CD86*) effects is the result of downstream signaling via MAP3K8 [25].

Both *TLR7* and *TLR3* are on the membrane of the endoplasmic reticulum (ER), nucleic acid-sensitive, and recruited from the ER to endolysosomes in response to stimulation by their ligands. It has been suggested that extracellular or endosomal DNases break down nucleic acids before TLRs recognize them.

*TLR3* is associated with the release of inflammatory cytokines such as *CCL2*, *CCL4*, *CCL5*, *CCL8*, and *CXCL10* after ds RNA stimulation, whereas *TLR7* signaling stimulates type I IFN production [26]. Note that *PRRX1*, a mesenchymal transcription factor present in Metagene4 (see Appendix A) but not selected by the pathway enrichment analysis, induces the gene for the TLR3 receptor.

An MES phenotype with a high basal inflammatory state is associated with the *PRRX1* gene in NB cells, which makes these cells capable of transcription for inflammatory signature genes and favors the recruitment of T cells (T-cell activation) and B cells (B-cell activation) [12]. Although *TLR7* signaling induces the production of type I IFN, genes belonging to the MHC-II (inducible by IFNɣ) are observed in the T-cell activation pathway.

*MHC-I* is generally expressed by all cells, yet tumor cells may lose its expression as a means of evading the immune system. Some malignancies are known to exhibit MHC-II even when they lose MHC-I expression; in our case, the *genes CD74*, *HLA-DMB*, *HLA-DPA1*, *HLA-DQA1*, *HLA-DQA2*, and *HLA-DQA2* were all expressed.

Recent studies suggest that *MHC-I* and *MHC-II* are independently regulated in different tumors [27,28]. The protein components of *MHC-II* are stabilized in the endoplasmic reticulum by the protein encoded by the *CD74* gene. This allows subsequent presentation of the *MHC-II-* restricted peptide on the cell surface. *CD74* regulates antigens presented by *MHC-II* and plays a dual role as a component of the *MHC-II* antigen presentation pathway and as a cytokine receptor. The expression of *CD74* in tumor cells is thought to play a pro- and anti-tumor role, which is due to its cytokine signaling and antigen presentation functions [29]. In NB cells, the presence of *MHC-II* may influence the immune response through mechanisms that allow direct interaction with T cells to induce polarization and activation. 

The binding of the proteins that make up the T cell/CD3 receptor complex (*CD247*, *CD3D*, *CD3E*, and *CD3G*), supported by *PTPRC* [30] and a second co-stimulating signal (*CD28*, *CD86*), provides the first signal for T-cell activation. 

*PTPRC* controls the action of the T cell/CD3 protein complex, which acts as a positive and negative regulator of the kinase LCK. The latter is dependent on cell type, *PTPRC* isoform expression, *PTPRC* inclusion or exclusion, and clustered signaling complexes. However, the *PTPRC* is linked to these responses as a key regulator of cytokine and chemokine gene expression [31]. These signals, which follow T/CD3-mediated phosphorylation of LCK, are directed to ZAP70 [32] to activate *VAV1* via LCP2 and work with WAS to control the cytoskeleton. 

Co-stimulation of *CD28*, which binds to *VAV1* and activates *RAC2*, is required for the second T-cell activation signal, which leads to the development of the AP1 (*FOS*, *JUNB*) transcription complex. Surface molecules transcribed by *CD28* and *CD86* are involved in the co-stimulation required to elicit T-cell responses. However, *CD28* deficiency can have an indirect effect on the B-cell compartment, leading to a lack of high-affinity antibodies and class switching due to insufficient helper T-cell proliferation. Therefore, *CD28* activation is essential for adaptive immunity [33]. By interfering with the control of inflammatory processes and tumor immunity, and by influencing the malignant development of cancer, alterations in the CD86 gene can inactivate T cells [34]. NB has been shown to have immunogenic effects in addition to T-lymphocyte and B-cell activation, which are both controlled by *PTPRC* and co-stimulated by the *CD19* complex, a co-receptor of the BCR B-cell receptor [35]. 

The initial stimulus allows *SYK* or a similar protein kinase linked to the *ZAP70* chain to be recruited and activated. *SYK*, a non-receptor-type protein kinase, is involved in the coupling of activated immune receptors; the resulting signaling model serves as a major paradigm of immune cell signaling. The stimulatory events generate a downstream signaling cascade involving *BTK*, *BLNK*, *VAV1*, and *RAC2*. Their transcriptional effects (*NFATC2*, *FOS*) lead to cytoskeletal regulation, proliferation, differentiation, and cytokine release [36]. This cascade is supported by *ITPR3* and *PIK3CG*.

This study found an interesting observation regarding the presence of several genes associated with inflammation, particularly the *VWF* gene. The *VWF* gene encodes a protein that plays a role in leukocyte adhesion, extravasation, and the release of pro-inflammatory cytokines. *VWF* acts as an adhesive surface on activated endothelial cells, facilitating the binding and retention of leukocytes. Tumor cells can use leukocyte-like trafficking pathways, including the acquisition of VWF expression. This suggests that NB tumor cells have an enhanced ability to interact with endothelial cells and penetrate the endothelium compared to other tumor cells that do not express VWF. As a result, both endothelial-cell-derived VWF and tumor-cell-derived VWF may contribute to tumor cell dissemination in the bloodstream [37]. The ability of tumor cells to communicate with endothelial cells via *VWF* is enhanced by the functions of the collagens present in the extracellular matrix (ECM), which help to establish a better interface with the microenvironment and make it permissive. Indeed, the kynurenine pathway in the inflammatory microenvironment [24] mutually potentiates the *COL12A1/ITGB* pathway [38] and plays a role in carcinogenesis and advanced clinical stage.

*COL12A1* and *COL14A1* are in the gene cluster for collagens that enrich the integrin signaling pathway; together with *COL15A1*, they belong to the family of FACIT collagens [39] to which the *COL6A2*, *COL6A3*, *COL5A1*, and *COL5A2* genes are related. Because they can adapt to environmental factors and assume varying degrees of stiffness, elasticity, and resistance, collagens, particularly FACITs, behave functionally. In fact, *COL5A1*, *COL5A2*, *COL6A2*, *COL6A3*, *COL12A1*, and *COL14A1* may be connected and have similar functions for the ECM [40]. The genes *COL5A3*, *COL1A2*, *COL4A5*, *COL8A1*, *COL8A2*, *COL10A1*, and *COL11A1*, which encode for collagens and are present in the integrin signaling pathway, make the interaction of NB cells with the microenvironment more effective; in fact, all these collagens have a function focused on organization, adhesion, regulation, and cell migration, ensuring the fluidity of the *ECM*. 

Signals from the *ECM* are involved in *PIK3* (*PIK3CG*, *PIK3R2*, *PIK3R5*) and activate second messengers (*RAC2*) that are critical in growth signaling pathways. These signals are mediated by *ITGB7* and *ITGAL*, *CCR1*, *CCR2*, *CCR5*, *CCR7*, *CX3CR1*, and *ITGAX* (inflammation mediated by chemokine and cytokine signaling pathway). *RAC2* plays an important role in many signaling pathways involving receptor tyrosine kinases, G-protein-coupled receptors (GPCRs), and integrin-related kinases [41]. The cytoskeletal rearrangements of actin (*ACTA2*, *ACTG2*, *FLNA*) are specifically initiated by VAV1 proteins as nucleotide exchange factors, and their action is ensured by *MYH11* and *MYH9*, contractile proteins that convert chemical energy into mechanical energy [42].

It was observed that *MYH11* truncating mutants exhibited increased motor activity and ATPase, suggesting a potential role for these proteins in energy balance and tumor cell motility [43]. Furthermore, *VAV1* results in the loss of intercellular adhesions and organized actin fibers, suggesting a link to invasiveness and EMT through the development of a mesenchymal phenotype. Axon guidance mediated by the netrin pathway (*NFATC2*, *PIK3CG*, *PIK3R2*, *PIK3R5*, and *RAC2*), whose genes have already been addressed, also modulates the invasive ability of NB cells. Conversely, netrin is a neuronal guidance signal that regulates the activation, migration, and cytoskeletal rearrangement of various cell types, including neurons. It is therefore a chemotropic protein whose signaling increases T-cell chemokinesis and promotes cell infiltration associated with acute inflammation in vivo [19].

Figure 2 illustrates all these signals and their “tumor-promoting inflammation” in NB. Finally, the inflammation pathway mediated by chemokines and cytokines is statistically the richest in genes (36 genes), confirming the crucial role played by the cellular protagonists of the immune system (T-cell activation; B-cell activation). By examining the genes enriched in the pathways selected by pathway enrichment analysis, some factors emerged that prevent NB from being classified as a “cold” or “non-immunogenic” tumor. These results allow the creation of an edit on which one can then develop a practical application to provide important feedback in clinical practice.

To obtain additional results and check what we have observed, we also reported the gene association networks obtained by the GeneMANIA software (version 3.5.2) [44] and the Search Tool for Retrieval of Interacting Genes (STRING) database (version 11.0) [45]. 

Both Neuroblastoma (CUI: C0027819) and Childhood Neuroblastoma (CUI: C4086165) were used as disease strings when we mined the DisGeNET database. We next filtered the list of linked genes using the subset of genes *PIK3CG*, *NFATC2*, *PIK3R2*, *VAV1*, *RAC2*, *COL6A2*, *COL6A3*, *COL12A1*, *COL14A1*, *ITGAL*, *ITGB7*, *FOS*, *PTGS2*, *PTPRC*, *ITPR3*. 

The eight genes *PIK3CG*, *PTGS2*, *FOS*, *PTPRC*, *VAV1*, *ITPR3*, *NFATC2*, and *PIK3R2* were identified as genes associated with the diseases in both cases (Neuroblastoma and Childhood Neuroblastoma disorders) with an EL_gda (Evidence Level gene-disease association) value larger than 0.6. Figure 3 depicts the gene association network obtained by GeneMANIA, while Figure 4 illustrates the network obtained by STRING when the list of these eight genes is entered as input.

These evaluations with the GeneMANIA and STRING genetic association networks have further strengthened the validity of the genes *PIK3CG*, *PTGS2*, *FOS*, *PTPRC*, *VAV1*, *ITPR3*, *NFATC2*, and *PIK3R2* in the pathogenesis of NB. Additionally, as Figure 3 and Figure 4 illustrate, these genes function as hubs, meaning that they are at the center of a network that connects them to each other as well as to other genes whose roles are related to the manifestation of the MES phenotype on an inflammatory basis.

## 3. Materials and Methods

This paper presents a retrospective study of fully anonymized NB samples from primary tumors of untreated patients. All of the data have been deposited in the NCBI Gene Expression database (https://www.ncbi.nlm.nih.gov/geo, accessed on 20 April 2022) [46] and contain the gene expression profiles of primary tumors; their identification is accomplished by entering keywords in the identification of the series (GSE) that belong to “Homo Sapiens” and by accurately screening the material to be chosen for the analyses inherent in the study. Data with the accession number GSE16476 were specifically chosen (https://www.ncbi.nlm.nih.gov/geo/query/acc.cgi?acc=GSE16476, accessed on 20 April 2022). 

The GSE16476 dataset presents 88 primary NB samples from untreated patients; the data in the GSE16476 dataset are devoid of all references regarding sex, age, and stage and refer to untreated primary neuroblastoma patients, whose profile was obtained from the residual material acquired during surgical interventions for diagnostic purposes and immediately frozen in liquid nitrogen. These specimens ranged from Neuroblastoma #1 to Neuroblastoma #88 (a detailed description of clinical and pathological data on the study population can be found in [47] and its related Appendix A; all data were fully anonymized). Gene profile was determined using the GPL570 platform Affymetrix Human Genome U133 Plus 2.0 Array. This platform represents a reference and guarantees complete coverage of the human U133 genome set with more than 6500 additional genes for analysis and more than 47,000 transcripts. All probe sets represented on the GeneChip Human Genome U133 set are identically replicated on the GeneChip Human Genome U133 Plus 2.0 array. The platform with the related protocol was published in 2003 and updated in 2020.

### Non-Negative Matrix Factorization for Automatic Gene Extraction

We analyzed these selected raw data using the non-negative matrix factorization (NMF) framework proposed in [48]. This framework was previously applied to identify low-dimensional structures embedded in colorectal cancer data, MCF-7 breast cancer data, and heterogeneous cancer-associated fibroblast population data [49,50,51]. This framework is based on a non-negative matrix factorization (NMF) algorithm which performs feature extraction on the non-negativity gene expression data matrix X ∈ R^n×m^ (representing the structured form—rows are genes and columns are samples—of the pre-processed data), decomposing it into additive combinations of two non-negative intrinsic feature factors: W ∈ R^n×k^ and H ∈ R^k×m^, such that X ≈ WH, with k < min (m,n). In particular, the metagene matrix W collects in its columns the biological information about the genes studied, while the elements of the coefficient matrix H show the importance of a particular metagene in each sample. Because of the non-negativity constraint, the NMF framework can interpret huge and complex data structures while preserving physical viability. It can also automatically identify the low-dimensional structure embedded in the original data. 

The NMF framework (sketched in Figure 5) has three major steps: (1) data pre-processing and matrix representation, (2) NMF core module, (3) metagene and common gene extraction. The data pre-processing and representation step begins with pre-processing the downloaded database to handle probe sets which refer to different quantities of transcript and associate probes that were not annotated in any sequence. This step produces the gene expression non-negative matrix X ∈ R^n×m^ that will be analyzed by the NMF core module. The second step, namely the NMF core module, computes the metagene and coefficient factor matrices W and H via sequential approximation of the Kullback–Leibler divergence loss optimizing the appropriate update rules (see [48,52] for mathematical details). Employing a gene score scheme suggested in [53], the third and final stage involves identifying the metagenes and extracting the relevant genes for each metagene.

In particular, the gene scoring computes a score value for each gene in a metagene and then selects only those genes overcoming a given threshold τ (the adopted threshold is $\tau =\;\hat{\mu }+3\;\hat{\sigma }$, with $\;\hat{\mu }$ and $\;\hat{\sigma }$ being the median and the MAD of the gene score vector itself). Metagenes in W containing the largest number of genes satisfying this empirical criterion are selected as the most representative of the information hidden in the microarray data matrix X. Specifically, in this study, five metagenes were estimated (rank k = 5 as illustrated in Figure 6). Metagene4 (presenting relevant genes) was extracted as the most representative element. Figure 7 illustrates the heatmap of the expression profiles of the extracted genes in Metagene4. This metagene was then used to perform the final pathway enrichment analysis as discussed in the Section 2.

## 4. Conclusions

In this review, we discussed the role of some genes within specific pathways in NB and how these genes are involved in the inflammatory process. In conclusion, we believe we have contributed to clarifying some aspects that characterize the inflammatory process in NB. Indeed, the specific framework used on NB data allowed the automatic extraction of some genes, whose analysis revealed their involvement in various inflammatory processes. The analyzed NB samples could secrete pro-inflammatory cytokines (*CCL2*, *CCL4*, *CCL5*, *CCL8*, *CXCL10*, etc.); produce PTGS2, which induces the synthesis of prostaglandins (*PGE2*); and modulate the microenvironment by interacting with collagens (*COL6A2*, *COL6A3*, *COL12A1*, *COL14A1*, *COL5A1*, *COL5A2*, etc.), which maintain and regulate the fluidity of the extracellular matrix (ECM). In addition, the samples could activate T cells (*CD247*, *CD3D*, *CD3E*, and *CD3G*) and B cells (*CD19*, *SYK*, *BTK*, and *BLNK*) and express signatures associated with immunotherapy. The idea that NB is unlikely to be classified as a non-immunogenic and/or “cold” tumor, and that it is only a stage-related disease at the time of diagnosis, is supported by the gene signals automatically retrieved from the existing NB data [54]. Numerous genes involved in inflammation, mediated by the integrin signaling pathway and the chemokine and cytokine signaling pathway, also characterize the MES phenotype, whose responses are strongly triggered by cues from T- and B-lymphocyte infiltrating cells associated with a significant inflammatory state.

Through a number of crosstalk mechanisms, further evaluated with gene association networks, the activated cellular pathways can produce complex pleiotropic effects and the inflammatory response. Furthermore, it is possible that therapeutic treatment could be helpful for the MES status of NB (*PRRX1*, *NFATC2*, *RAC2*, *VAV1*), which showed a higher degree of inflammatory characterization and possible immunological vulnerability. A thorough evaluation combined with additional testing of the results of the analysis in this article may shed light on some of the characteristics of the inflammatory process in NB. However, these new insights into the phenotype, adaptability, and involvement of non-tumorigenic cells (T and B lymphocytes) within the tumor microenvironment have further reinforced the importance of disease stage (MES) in treatment selection. Advances in molecular biology and pathophysiology could make it possible to provide more specialized therapies for kids with NB. Those who are now incurable should benefit from this.

## Figures and Tables

**Figure 1 ijms-25-04406-f001:**
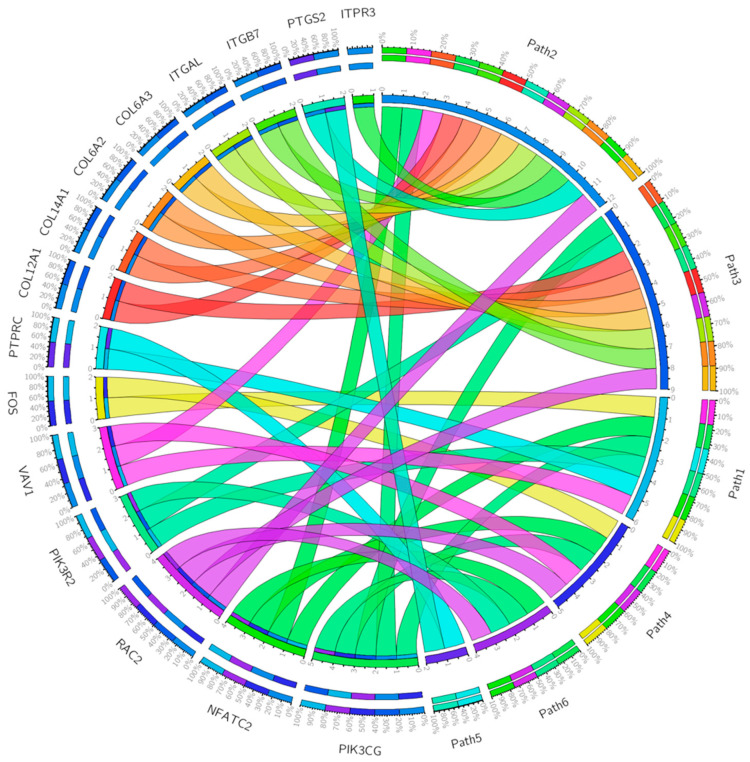
Shared genes in the six pathways selected by the WebGeastalt tool that meet the significance criteria (FDR < 1; *p*-value < 0.05). The circular graph showing the binary relationship between the six selected pathways and their shared genes was obtained using Circos visualization software [14] (Version 3, 29 June 2007). The ribbons represent the relationships (the twists on the ribbons indicate the orientation of the link), the circular segments represent the genes and the pathways to which they are linked (Path1 = T-cell activation, Path2 = chemokine- and cytokine-mediated inflammation pathway, Path3 = integrin signaling pathway, Path4 = B-cell activation, Path5 = Toll receptor signaling pathway, Path6 = netrin-mediated axon guidance). Inflammation mediated by the chemokine and cytokine signaling pathway shares 12 genes: *COL6A2*, *COL6A3*, *COL12A1*, *COL14A1*, *ITGB7*, *ITGAL*, *ITPR3*, *NFATC2*, *PIK3CG*, *PTGS2*, *RAC2*, *VAV1*. The integrin signaling pathway shares 9 genes: *COL6A2*, *COL6A3*, *COL12A1*, *COL14A1*, *ITGAL*, *ITGB7*, *PIK3CG*, *PIK3R2*, *RAC2*. *B-cell activation shares 6 genes: ITPR3*, *FOS*, *NFATC2*, *PIK3CG*, *PTPRC*, *RAC2*, *VAV1*. T-cell activation shares 5 genes: *FOS*, *NFATC2*, *PIK3R2*, *PIK3CG*, *PTPRC*, *VAV1*. Netrin-mediated axon guidance shares 4 genes: *NFATC2*, *PIK3CG*, *PIK3R2*, *RAC2*. The toll receptor signaling pathway shares only the gene *PTGS2*.

**Figure 2 ijms-25-04406-f002:**
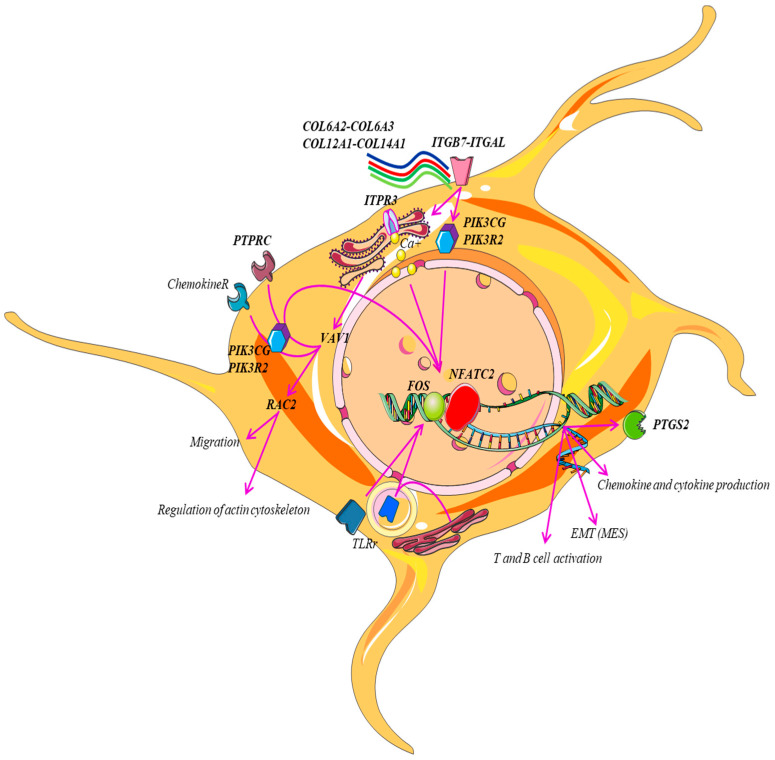
Representative scheme of the 15 genes shared by the selected pathways using pathway enrichment analysis. Our model shows the mechanisms that converge to activate *NFATC2* and other targets such as *PIK3CG* that are shared by the action of different stimuli. Some factors in the tumor microenvironment (*COL6A2*, *COL6A3*, *COL12A1*, *COL14A1*, chemokines, cytokines, etc.) can promote crosstalk between *NFATC2* and *PIK3CG* and can promote crosstalk between receptor types (*ITGB7*, *ITGAL*, *PTPRC*, chemokine R) that induce the activation of signals (*ITPR3*, *PIK3CG*, *PIK3R2*, *VAV1*, *RAC2*) that have downstream pro-inflammatory and non-pro-inflammatory effects: production of *PTGS2*, cytokines, and chemokines and activation of T and B cells.

**Figure 3 ijms-25-04406-f003:**
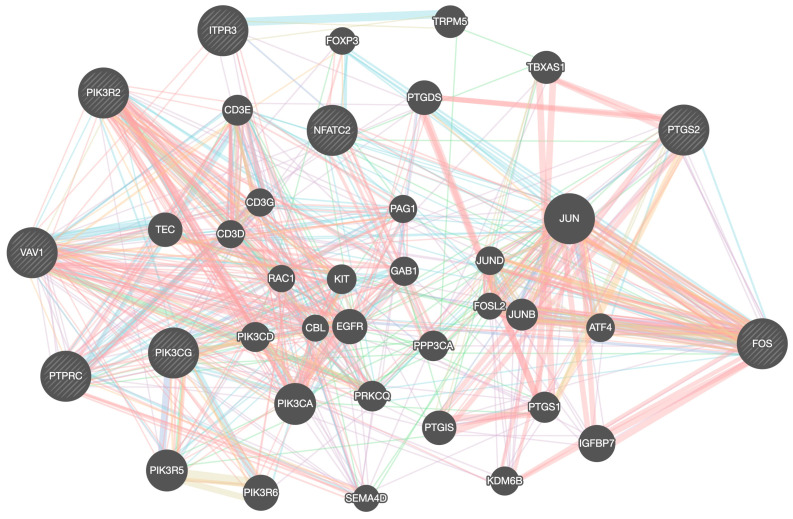
The gene association networks that result from entering the list of eight genes—*PIK3CG*, *PTGS2*, *FOS*, *PTPRC*, *VAV1*, *ITPR3*, *NFATC2*, and *PIK3R2*—as input into the GeneMANIA software. GeneMANIA extends the user’s list with genes that are functionally similar, or share properties with the initial query genes, and displays an interactive functional association network, illustrating the relationships among these genes and datasets. The size of each node is proportional to the number of physical interactions (pink edges) among other genes and indicates the importance of this gene as a hub in the network. Genes that are co-expressed are shown by light violet edges, and pathways involving related nodes are indicated by cyan edges.

**Figure 4 ijms-25-04406-f004:**
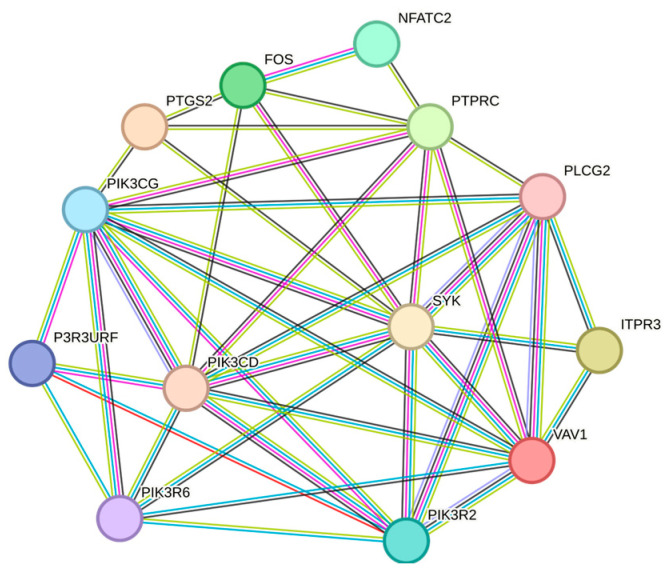
The gene association networks that result from entering the list of eight genes—*PIK3CG*, *PTGS2*, *FOS*, *PTPRC*, *VAV1*, *ITPR3*, *NFATC2*, and *PIK3R2*—as input into the STRING software (version 3.5.2). The depicted network has 13 nodes, 41 edges, an average node degree of 6.31, and a PPI enrichment *p*-value of 7.59 × 10^−13^.

**Figure 5 ijms-25-04406-f005:**
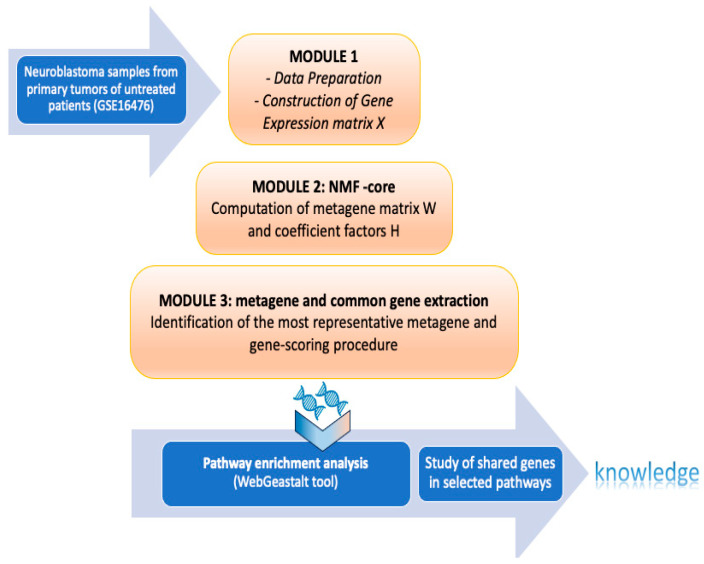
Scheme of the NMF framework adopted to extract subgroups of genes relevant to the sample of NB included in the GSE16476 dataset. The three sequential steps are as follows: (1) pre-process raw data and construct a matrix representation of them, namely the gene expression non-negative matrix X; (2) factorize X into metagene matrix W and a coefficient factor H (with rank information equal to r = 5); (3) identify the most representative metagene in W, namely Metagene4, and rank its relevant genes. The subset of selected genes is subjected to post-processing analysis to determine their role in the inflammatory process and pathway enrichment analysis in NB.

**Figure 6 ijms-25-04406-f006:**
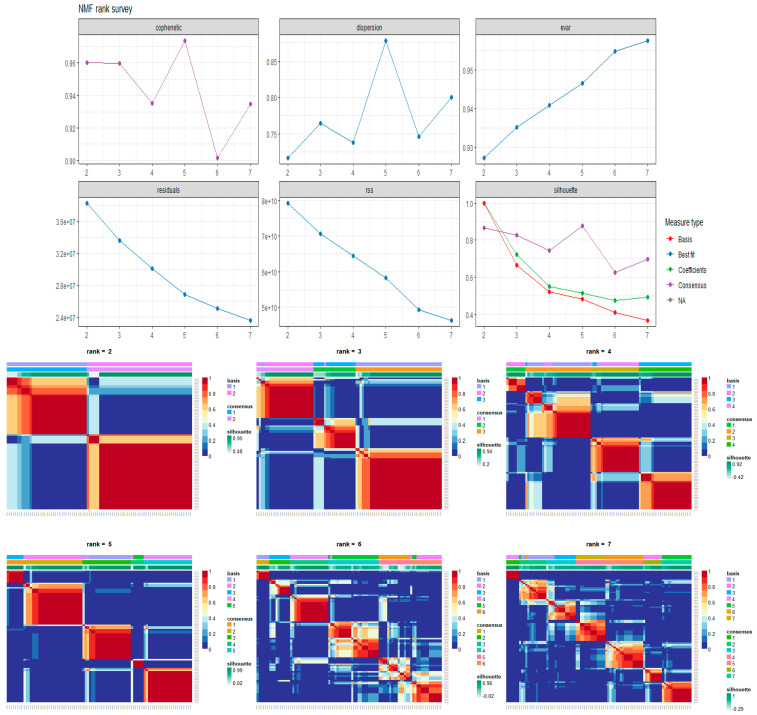
Evaluation of the most appropriate rank value to be used in the factorization process of the microarray matrix representing the NB data. The six pictures in the first panel (NMF rank survey) report the behavior of different measures useful to define the proper number of metagenes to extract from the data (i.e., the rank r of the factorization process). The second panel, instead, reports the plots of the consensus matrices obtained with different rank values (r = 2…,7). From a deep analysis of both the plots of the Cophenetic Correlation Coefficients (CCC) and the consensus matrices, it can be deduced that the value k = 5 is the first value at which the CCC trend starts to decrease. This made it possible to determine the rank of k = 5 for the purpose of computing the matrices W and H, that are the most appropriate to discretize the NB dataset.

**Figure 7 ijms-25-04406-f007:**
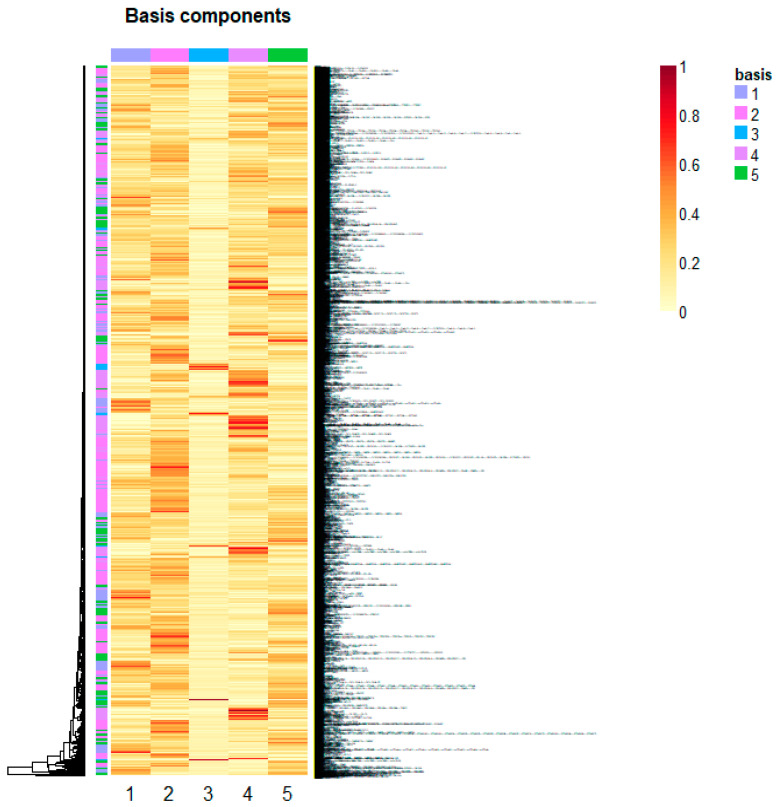
Expression profiles of the extracted genes plotted as a heatmap. Rows show individual genes, columns show individual samples. Red tiles indicate expression above, yellow tiles indicate expression below the median transcript level for that gene across all samples. Color saturation is proportional to the magnitude of the difference from the median. Genes and conditions were clustered using an average linkage algorithm.

**Table 1 ijms-25-04406-t001:** Pathway enrichment analysis: Ten categories are identified as enriched categories, and all are shown in this table. The parameters for the enrichment analysis are as follows: minimum number of IDs in the category: 5, maximum number of IDs in the category: 2000, FDR method: BH, significance level: top 10.

Gene Set	Description	Size	Expect	Ratio	*p* Value	FDR
P00053	T-cell activation	75	4.8985	4.4912	4.9299 × 10^−10^	5.5708 × 10^−8^
P00031	Inflammation mediated by chemokine and cytokine signaling pathway	200	13.063	2.7559	3.6630 × 10^−9^	2.0696 × 10^−7^
P00034	Integrin signaling pathway	166	10.842	2.1214	0.00030327	0.011423
P00010	B-cell activation	58	3.7882	2.9038	0.00099971	0.028242
P00054	Toll receptor signaling pathway	50	3.2657	2.1435	0.041004	0.79098
P00009	Axon guidance mediated by netrin	30	1.9594	2.5518	0.041999	0.79098
P00050	Plasminogen activation cascade	15	0.97970	3.0622	0.069606	1
P00033	Insulin/IGF pathway–protein kinase B signaling cascade	35	2.2860	2.1873	0.073811	1
P00011	Blood coagulation	38	2.4819	2.0146	0.097796	1
P00047	PDGF signaling pathway	125	8.1642	1.4698	0.11051	1

**Table 2 ijms-25-04406-t002:** List of genes present in the first 6 pathways selected with the WebGeastalt tool and satisfying the significance criteria (FDR < 1; *p*-value < 0.05). The parameters for the enrichment analysis are as follows: minimum number of IDs in the category: 5, maximum number of IDs in the category: 2000, FDR method: BH, significance level: top 10.

*T-cell activation*	*CD247*, *CD28*, *CD3D*, *CD3E*, *CD3G*, *CD74*, *CD86*, *FOS*, *HLA-DMB*, *HLA-DPA1*, *HLA-DQA1*, *HLA-DQA2*, *HLA-DRA*, *LCK*, *LCP2*, *NFATC2*, *PIK3CG*, *PIK3R2*, *PTPRC*, *VAV1*, *WAS*, *ZAP70*
*Inflammation mediated by chemokine and cytokine signaling pathway*	*ACTA2*, *ACTG2*, *ALOX5AP*, *C5AR1*, *CCL11*, *CCL13*, *CCL18*, *CCL2*, *CCL21*, *CCL4*, *CCL5*, *CCL8*, *CCR1*, *CCR2*, *CCR5*, *CCR7*, *COL12A1*, *COL14A1 COL6A2*, *COL6A2*, *COL6A3*, *CX3CR1*, *CXCL10*, *FPR1*, *FPR3*, *ITGAL*, *ITGB7*, *ITPR3*, *JUNB*, *MYH11*, *MYH9*, *NFATC2*, *PIK3CG*, *PTGS2*, *RAC2*, *VAV1*, *VWF*
*Integrin signaling pathway*	*COL10A1*, *COL11A1 COL12A1*, *COL14A1*, *COL15A1*, *COL1A2*, *COL4A5*, *COL5A1*, *COL5A2*, *COL5A2*, *COL6A2*, *COL6A3*, *COL8A1*, *COL8A1*, *FLNA*, *ITGAL*, *ITGAX*, *ITGAX*, *ITGB7*, *ITGBL1*, *PIK3CG*, *PIK3R2*, *PTGS2*, *RAC2*
*B-cell activation*	*BLNK*, *BTK*, *CD19*, *FOS*, *ITPR3*, *NFATC2*, *PIK3CG*, *PTPRC*, *RAC2*, *SYK*, *VAV1*
*Toll receptor signaling pathway*	*CD14*, *LY96*, *MAP3K8*, *PTGS2*, *TLR2*, *TLR3*, *TLR7*
*Axon guidance mediated by netrin*	*NFATC2*, *PIK3CG*, *PIK3R2*, *PIK3R5*, *RAC2*

## Data Availability

Fully anonymized gene expression profiles of primary NB tumors of untreated patients. All the data were downloaded from the NCBI Gene Expression database (https://www.ncbi.nlm.nih.gov/geo (accessed on 20 April 2022)) as described in the Section 3.

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
