# Peer review of "Review of Patient Gene Profiles Obtained through a Non-Negative Matrix Factorization-Based Framework to Determine the Role Inflammation Plays in Neuroblastoma Pathogenesis"

_ijms, 2024, doi:10.3390/ijms25084406_

Round 1

Reviewer 1 Report

Comments and Suggestions for Authors

This study used neuroblastoma samples from untreated patients and analyzed 88 gene profiles through a Nonnegative Matrix Factorization framework. The identified subgroup of genes sheds light on the inflammatory phenotype of neuroblastoma. However, despite these findings, there are major concerns regarding the organization, clarity, and data presentation in the paper that should to be addressed before its publication.

The manuscript requires deep proofreading and editing to improve structure and the flow of the text.

The abstract is a bit too straightforward and should clearly highlight the contribution of this study to the field.

The Introduction presents a lot of information without the appropriate bibliographic references. Citations should accompany the information being provided in each statement/paragraph. 

The final paragraph of the introduction is unnecessary and drives attention away from the objective of the study.

How do the genomic alterations described in the introduction affect tumor behavior or patient prognosis? And potentially inflammatory pathways?

The introduction discusses the discovery of two epigenetic states in neuroblastoma. Can the authors elaborate on the implications of these states for tumor behavior, treatment response, and patient outcomes?

Why was this specific dataset, GSE16476, chosen?

The description of the three major steps of the NMF framework is detailed and provides a clear understanding of the analytical process. However, the authors should consider breaking down each step into smaller paragraphs for enhanced readability.

Table 2 is not at all reader-friendly, the authors should consider editing it to avoid the blank boxes.

Presentation of results is lacking a bit of clarity, the figures themselves are a bit confusing and the figure captions do not entirely help to clarify them.

Why was the PAN-THER database chosen as the reference database for clustering the genes into functional pathways? What advantages does PAN-THER offer for this analysis?

Could you provide more details on the methodology used for pathway enrichment analysis, including any specific parameters or thresholds applied during the analysis?

Were there any limitations or potential biases in the selection of primary neuroblastoma tumor profiles for gene analysis? If so, how were these addressed or accounted for in the study?

In the discussion of shared genes among pathways, did the authors investigate potential interactions or crosstalk between these genes within pathways? 

Considering the complex nature of tumor microenvironment, were any additional factors or variables considered in the analysis of gene pathways and their implications for neuroblastoma progression?

Considering the heterogeneity of neuroblastoma tumors, did the authors conduct any subgroup analyses or correlations between gene pathways and clinical outcomes? 

Study limitations should be clearly addressed in the discussion of the paper. 

Comments on the Quality of English Language

The quality of english language in the manuscript requires significant improvement. There are grammatical errors, awkward phrasings, and unclear expressions throughout the text. Sentence structure and word choice should also be a concern to ensure readability and comprehension. 

Author Response

Dear Reviewer #1,

please find enclosed the revised version of the manuscript entitled “Review of patient gene profiles obtained through an NMF-based framework to determine the role inflammation plays in neuroblastoma pathogenesis” which we are re-submitting for a possible publication in International Journal of Molecular Science.

The manuscript has benefited from a complete rewording and a thorough revision of all sections to correct typos and to improve the readability of the contents.  All the specific changes have been evidenced in blue color in the resubmitted version. In response to the reviewers' observations, some materials have been added or reformulated.

A detailed reply to your observation has been also reported below.

Reviewer #1: This study used neuroblastoma samples from untreated patients and analyzed 88 gene profiles through a Nonnegative Matrix Factorization framework. The identified subgroup of genes sheds light on the inflammatory phenotype of neuroblastoma. However, despite these findings, there are major concerns regarding the organization, clarity, and data presentation in the paper that should be addressed before its publication.

The manuscript requires deep proofreading and editing to improve structure and the flow of the text.

The abstract is a bit too straightforward and should clearly highlight the contribution of this study to the field. The Introduction presents a lot of information without the appropriate bibliographic references. Citations should accompany the information being provided in each statement/paragraph. The final paragraph of the introduction is unnecessary and drives attention away from the objective of the study.

Reply: We appreciate the reviewer's comments on the manuscript and her thorough reading of it. Regarding these observations we improved the abstract to clarify the purpose of this study. All the bibliographical references have been checked and some additional recent references were appropriately cited.

Regarding the comment on the last paragraph, it may appear distracting, but we preferred to maintain it int this revision since it defines the editing of the work giving the reader a brief road map of all the work contents.

How do the genomic alterations described in the introduction affect tumor behavior or patient prognosis? And potentially inflammatory pathways?

Reply: We appreciate the reviewer's comment and would like to point out that the data emerging from the literature are mainly aimed at identifying the molecular mechanisms involved in the pathogenesis of neuroblastoma, as very few gene defects are present. Furthermore, these products have selected several signaling pathways necessary for the growth and progression of neuroblastoma and some also contribute to resistance to conventional treatments of the disease (see reference [5]).

The introduction discusses the discovery of two epigenetic states in neuroblastoma. Can the authors elaborate on the implications of these states for tumor behavior, treatment response, and patient outcomes?

Reply: The recent identification of the two different epigenetic states, mesenchymal (MES) and adrenergic (ADRN), in neuroblastoma has initiated a series of experimental approaches that obviously require a better understanding of the molecular factors that control phenotypic plasticity, but above all the availability of a larger number of patients. All this information can be obtained with the general participation of many colleagues focused on this topic and not only from the discussion addressed in this paper. A clearer picture of this topic will soon be a fundamental step in the development of more effective therapies that improve the outcome of neuroblastoma patients.

Why was this specific dataset, GSE16476, chosen?

Reply: The choice of the GSE16476 dataset is motivated by: i) the presence of 88 primary neuroblastoma samples from untreated patients, necessary for the purpose of our study, i.e. all samples must not have received any treatment. Added to this is the difficulty of having datasets with a high number of samples given the small number of patients. The neuroblastoma samples in this dataset constitute residual material obtained during surgical interventions for diagnostic purposes and immediately frozen in liquid nitrogen. The data have been deposited in the NCBI Gene Expression Omnibus (http://www.ncbi.nlm.nih.gov/geo/) whose accession number is GSE16476 [Lamers F, Schild L, Koster J, Speleman F et al. Identification of BIRC6 as a novel intervention target for neuroblastoma therapy. BMC Cancer 2012 Jul 12; 12:285. PMID: 22788920]. Ethical approval from the Dutch Medical Ethics Committee was not required for the use of the surplus materials in the Netherlands. However, informed consent was obtained from the patients' parents for the use of this material, which is archived at the University of Amsterdam Academic Medical Center. ii) from transcriptomic analysis with the Affymetrix GPL570 platform. This platform represents a reference and guarantees complete coverage of the human U133 genome set with more than 6,500 additional genes for analysis and more than 47,000 transcripts. All probe sets represented on the GeneChip Human Genome U133 set are identically replicated on the GeneChip Human Genome U133 Plus 2.0 array. The platform with the related protocol was published in 2003 and updated in 2020.

The description of the three major steps of the NMF framework is detailed and provides a clear understanding of the analytical process. However, the authors should consider breaking down each step into smaller paragraphs for enhanced readability.

Table 2 is not at all reader-friendly, the authors should consider editing it to avoid the blank boxes.

Reply: Table 2 has been reorganized, for the sake of clearness and readability. Additional spaces were removed as suggested.

Presentation of results is lacking a bit of clarity, the figures themselves are a bit confusing and the figure captions do not entirely help to clarify them.

Reply: Additional figure describing the NMF-based approach has been added. The figure captions were better described for the for the sake of clearness and readability.

Why was the PANTHER database chosen as the reference database for clustering the genes into functional pathways? What advantages does PANTHER offer for this analysis?

Could you provide more details on the methodology used for pathway enrichment analysis, including any specific parameters or thresholds applied during the analysis?

Were there any limitations or potential biases in the selection of primary neuroblastoma tumor profiles for gene analysis? If so, how were these addressed or accounted for in the study?

Reply: The PANTHER database is present in the WebGestalt tool (WEB-based Gene SeT AnaLysis Toolkit) (see reference [21]). This tool is a functional enrichment analysis web tool. According to Google Analytics, it has an average of 26,000 unique users from 144 countries and more than 2,500 citations in scientific articles per year. WebGestalt supports three well-established and complementary methods for enrichment analysis, including over-representation analysis (ORA), gene set enrichment analysis (GSEA), and network topology-based analysis (NTA). The PANTHER classification system (http://www.pantherdb.org) is a comprehensive system that combines genomes, gene function classifications, pathways, and statistical analysis tools to enable biologists to analyze large-scale experimental data at the entire genome. The current PANTHER system v.14.0 (Huaiyu Mi, Anushya Muruganujan, Xiaosong Huang, Dustin Ebert, Caitlin Mills, Xinyu Guo, Paul D Thomas: Protocol Update for large-scale genome and gene function analysis with the PANTHER classification system (v.14.0 ); Nat Protoc. 2019 Mar;14(3):703-721. doi: 10.1038/s41596-019-0128-8. Epub 2019 Feb 25) covers 131 complete genomes organized into families and subfamilies of genes; evolutionary relationships between genes are represented in phylogenetic trees, multiple sequence alignments, and statistical models. Path analysis, in WebGestalt, can also be carried out with KEGG but the statistical resolution, in our study, was more stringent with PANTHER; furthermore, routes that are strongly implicated in the topic discussed were selected. In our study we use the ORA enrichment analysis which provides a single column with the output genes obtained from the NMF analysis. These genes are analyzed according to the following parameters: the reference organism: hsapiens; enrichment categories: Panther pathways; the type ID: gene-symbol; the reference list: genome-protein-coding.

Furthermore, the enrichment analysis was set with a minimum and maximum number of IDs.

The statistical method chosen for the FDR is BH (Yoav Benjamini and Yosef Hochberg BH procedure). The significance level used is required for the top 10 pathways.

In the discussion of shared genes among pathways, did the authors investigate potential interactions or crosstalk between these genes within pathways?

Reply: We appreciate the reviewer's comment to clarify the issue.  As shown in the results, some genes are shared in multiple pathways, just read what was written from line 190 to 194: "PIK3CG is present in 5/6 pathways, the NFATC2 gene is present in 4/6 pathways, the genes PIK3R2, VAV1 and RAC2 are present in 3/6 pathways, finally the genes COL6A2, COL6A3, COL12A1, COL14A1, ITGAL, ITGB7, FOS, PTGS2, PTPRC and ITPR3 are shared in 2/6 pathways". Sharing of genes is a recurring fact in biology, and this highlights how some genes can have a pleiotropic role.

Considering the complex nature of tumor microenvironment, were any additional factors or variables considered in the analysis of gene pathways and their implications for neuroblastoma progression?

Reply: The role of the microenvironment is certainly relevant in tumor progression (see references [17, 18]). The discussion of the results for each gene selected highlights the importance of the communication network established between the tumor parenchyma and the stroma, which contains numerous cellular elements (fibroblasts, macrophages, lymphocytes, etc.). When analyzing the results obtained, we detected the presence of genes coding for inflammatory cytokines (CCL2, CCL4, CCL5, CCL8, CXCL10, etc.) and the production of PTGS2 (responsible for the synthesis of PGE2); furthermore, the interaction with the microenvironment is also modulated with a crosstalk through the synthesis of collagens (COL6A2, COL6A3, COL12A1, COL14A1, COL5A1, COL5A2, etc.). Of course, it is only by referring to these data, in addition to what has been discussed in the experimental work, that one can understand how the microenvironment, in this case the inflammatory one, is responsible for the progression of the tumor.

Considering the heterogeneity of neuroblastoma tumors, did the authors conduct any subgroup analyses or correlations between gene pathways and clinical outcomes?

Reply: We appreciate the reviewer's question that is certainly pertinent and we would like to point out that represents an objective of future application studies that will involve the use of the most representative genes.

Study limitations should be clearly addressed in the discussion of the paper.

Reply: The final discussion section has been reorganized to demonstrate the role of some genes within specific pathways in NB and how these genes are involved in the inflammatory process and the contribution of this study to clarify some aspects that characterize the inflammatory process in NB. 

Comments on the Quality of English Language

The quality of English language in the manuscript requires significant improvement. There are grammatical errors, awkward phrasings, and unclear expressions throughout the text. Sentence structure and word choice should also be a concern to ensure readability and comprehension.

Reply: The manuscript has benefited from a complete rewording and a thorough revision of all sections to fix typos, correct grammatical errors, and to improve the readability of the paper. The English presentation has been checked and improved.

Reviewer 2 Report

Comments and Suggestions for Authors

In the current study, Boccarelli and colleagues identified key inflammatory-related genes associated with neuroblastoma pathogenesis using a non-negative matrix factorization framework. The obtained results can be considered as novel data that reject the hypothesis of an immunologically cold type of neuroblastoma and provide valuable information for the development of novel approaches in neuroblastoma treatment. Despite obtaining highly interesting results based on deep bioinformatic analysis of transcriptomic data, the information contained in this manuscript is insufficient for successful publication in Int. J. Mol. Sci.

The main comment on this paper is the lack of verification of the data obtained in independent samples and experimental models. Indeed, the identified genes play a key role in oncogenesis according to the literature, as detailed by the authors in paragraph 4, but the importance of such genes in neuroblastoma pathogenesis needs to be confirmed by independent methods (e.g. expression levels in different neuroblastoma GEO datasets, association with neuroblastoma patient survival, hub positions of the revealed genes in neuroblastoma-related gene association networks, changes in expression levels of these genes when neuroblastoma growth is enhanced in experimental models, etc.). In its present form, the results obtained are unfortunately illusory. In my opinion, this manuscript cannot be accepted for publication in Int. J. Mol. Sci. The authors should revise their work and resubmit it to this journal (perhaps Genes from MDPI is a more appropriate journal for this publication).

Minor comments:

1. Please correct the figures so that all legends and captions are visible.

2. Please provide a more detailed description of the methodology used to identify key genes.

3. Please reformat references according to the rules for authors (e.g. references cited in lines 94, 105, 109, 111, 135, 141, etc.).

Author Response

Dear Reviewer #2,

please find enclosed the revised version of the manuscript entitled “Review of patient gene profiles obtained through an NMF-based framework to determine the role inflammation plays in neuroblastoma pathogenesis” which we are re-submitting for a possible publication in International Journal of Molecular Science.

The manuscript has benefited from a complete rewording and a thorough revision of all sections to correct typos and to improve the readability of the contents.  All the specific changes have been evidenced in blue color in the resubmitted version. In response to the reviewers' observations, some materials have been added or reformulated.

A detailed reply to your observation has been also reported below.

Reviewer #2: In the current study, Boccarelli and colleagues identified key inflammatory-related genes associated with neuroblastoma pathogenesis using a non-negative matrix factorization framework. The obtained results can be considered as novel data that reject the hypothesis of an immunologically cold type of neuroblastoma and provide valuable information for the development of novel approaches in neuroblastoma treatment. Despite obtaining highly interesting results based on deep bioinformatic analysis of transcriptomic data, the information contained in this manuscript is insufficient for successful publication in Int. J. Mol. Sci.

The main comment on this paper is the lack of verification of the data obtained in independent samples and experimental models. Indeed, the identified genes play a key role in oncogenesis according to the literature, as detailed by the authors in paragraph 4, but the importance of such genes in neuroblastoma pathogenesis needs to be confirmed by independent methods (e.g. expression levels in different neuroblastoma GEO datasets, association with neuroblastoma patient survival, hub positions of the revealed genes in neuroblastoma-related gene association networks, changes in expression levels of these genes when neuroblastoma growth is enhanced in experimental models, etc.). In its present form, the results obtained are unfortunately illusory. In my opinion, this manuscript cannot be accepted for publication in Int. J. Mol. Sci. The authors should revise their work and resubmit it to this journal (perhaps Genes from MDPI is a more appropriate journal for this publication).

Reply: We appreciate the reviewer's comments on the manuscript and her thorough reading of it. Regarding her comments, these are certainly relevant, however this represents a target for future applied studies.  In fact, the selected genes will be verified using appropriate molecular and cellular biology techniques and will be used to design and select new drugs with the aim of obtaining more effective therapies to improve the outcome of neuroblastoma patients. What has been done with this experimental work is to create an editing on which to subsequently develop the practical application.
We would like also to emphasize that manuscript was reworded, some materials have been added and the following referee observations are discussed in detail:

Minor comments:

  1. Please correct the figures so that all legends and captions are visible.

Reply: All the captions and legends were adjusted.

  1. Please provide a more detailed description of the methodology used to identify key genes.

Reply: Some details about the NMF-based methodology used to automatically extract the key genes were added. The adopted framework is described in all this mathematical and implemented issues in the reference [15] doi: 10.1177/1177932220906827. A novel figure (Figure 1) illustrating the  scheme of the NMF-framework adopted to analyze samples of NB included in the GSE16476 dataset has been added, hoping this can help the reader.

  1. Please reformat references according to the rules for authors (e.g. references cited in lines 94, 105, 109, 111, 135, 141, etc.).

Reply: All references were checked and formatted according to the journal rule (…References may be in any style, provided that you use the consistent formatting throughout. It is essential to include author(s) name(s), journal or book title, article, or chapter title (where required), year of publication, volume and issue (where appropriate) and pagination. DOI numbers (Digital Object Identifier) are not mandatory but highly encouraged.)

Reviewer 3 Report

Comments and Suggestions for Authors

Dear Authors, I carefully read the manuscript that you submitted to IJMS. I think that the manuscript deals with an important topic that deserve more focused research, like yours. Nevertheless, I found some flaws that in my opinion need to be corrected prior the publication. 

First, a general comment: maybe you tried to summarize a lot of topics but in my opinion you summarized too much. I mean that the topic deserved more extensive explanations and I think that the Discussion section must be improved. In the present form, the Discussion seems to be a list of acronyms and a few explanations are provided to readers. And this leads to one of the major flaws: the correlation with clinical practice is completely missing. I think that your efforts will be better emphasized if you clearly explain all the clinical implications of your results. 

More precise comments are provided as follows:

1. What is the main question addressed by the research? Authors would like to expose to readers the results obtained by a retrospective gene profile analysis in patients affected by neuroblastoma.
2. What parts do you consider original or relevant for the field? What
specific gap in the field does the paper address? The originality of the manuscript is hard to identify. Authors never declare the originality of the proposed data. I encourage to rephrase some sentences in introduction to highlight the scope of the paper and also to include some sentences in Discussion in order to better clarify the originality of the manuscript. 
3. What does it add to the subject area compared with other published
material? Authors aimed to demonstrate the inflammatory etiology of neuroblastoma.
4. What specific improvements should the authors consider regarding the
methodology? What further controls should be considered? Authors wrote along M%M section that they collect gene profiles of anonymous patients. Nevertheless I think that significant information should be proposed to readers according to the neoplastic staging. Information about sex, age and other clinical information about patients are negligible but the tumor staging should be included since it can influence the results.
5. Please describe how the conclusions are or are not consistent with the
evidence and arguments presented. Please also indicate if all main questions
posed were addressed and by which specific experiments.

The conclusion section should be rephrased since, in the present form, it seems more the conclusive part of the Discussion section. 
6. Are the references appropriate?

Yes
7. Please include any additional comments on the tables and figures and
quality of the data.

Data, figures and graphs help the comprehension of the manuscript. 

L27- 30: rephrase: Neuroblastoma (NB) is an extracranial solid tumor, and it is considered the most frequent diagnosed  neoplasia in children (7% 29 - 10%) after leukemia and brain tumors [1,2].

L30: do not start a sentence with an abbreviation (the same also for AA at L265)

L73: according to the sentence proposed by the authors, it seems that MHC-1 is the only marker to define a "cold tumor" but then, according to the Discussion section, it is not. Please, provide a good introduction in order to guide the readers to through the text properly. 

L90: the aim of the study is missing.

L93: what authors mean with "untreated patients"? Have samples been collected prior surgery and chemo? Or patients never received any treatment? 

L91-106: declare the tumor staging here if possible. 

L 106: rephrase: ... materials). All data were fully anonymized.

L109: add a space prior (NMF)

L141: a sentence written in Italian is present. the same in L180-181

L154: (r=2…,7) is it correct? 

L213, 220, 222, 227, 250, 253: Axon and Inflammation must have a capital letter? 

L239 and L 270: correct  [23; 25] instead of  [23], [25].

L243: comma instead of full stop after B cell activation

L348: This suggests that NB tumour cells instead of "This suggests that neuroblastoma (NB) tumour cells"

L395: clinical implications? please, add some comments here. 

Comments on the Quality of English Language

English language requires minor editing. 

Please, take care to build properly the sentences according to English style (from time to time, the sentences have a Latin flavor).

Author Response

Dear Reviewer #3,

please find enclosed the revised version of the manuscript entitled “Review of patient gene profiles obtained through an NMF-based framework to determine the role inflammation plays in neuroblastoma pathogenesis” which we are re-submitting for a possible publication in International Journal of Molecular Science.

The manuscript has benefited from a complete rewording and a thorough revision of all sections to correct typos and to improve the readability of the contents.  All the specific changes have been evidenced in blue color in the resubmitted version. In response to the reviewers' observations, some materials have been added or reformulated.

A detailed reply to your observation has been also reported below.

Reviewer #3: Dear Authors, I carefully read the manuscript that you submitted to IJMS. I think that the manuscript deals with an important topic that deserve more focused research, like yours. Nevertheless, I found some flaws that in my opinion need to be corrected prior the publication. 
First, a general comment: maybe you tried to summarize a lot of topics but in my opinion you summarized too much. I mean that the topic deserved more extensive explanations and I think that the Discussion section must be improved. In the present form, the Discussion seems to be a list of acronyms and a few explanations are provided to readers. And this leads to one of the major flaws: the correlation with clinical practice is completely missing. I think that your efforts will be better emphasized if you clearly explain all the clinical implications of your results. 

More precise comments are provided as follows.
Reply: We thank the reviewer for her kind judgments on the manuscript and the careful reading of it. Regarding her comments, which were certainly appropriate, we would like to point out that we reviewed the entire manuscript, adjusting texts that caused misunderstandings, adding some new materials that we hope will help the reader better understand the obtained results, correct typos, and properly formatted all references. The following referee observations are discussed in detail:

  1. What is the main question addressed by the research? Authors would like to expose to readers the results obtained by a retrospective gene profile analysis in patients affected by neuroblastoma.
  2. What parts do you consider original or relevant for the field? What
    specific gap in the field does the paper address?
    The originality of the manuscript is hard to identify. Authors never declare the originality of the proposed data. I encourage to rephrase some sentences in introduction to highlight the scope of the paper and also to include some sentences in Discussion in order to better clarify the originality of the manuscript. 

Reply: We thank the reviewer for her suggestions, we rephrase some sentences in abstract, introduction, and discussion session to better clarify the aim of the research, the novelty of the NMF-based approach used to automatic extract latent information from the specific neuroblastoma dataset and the role of some of the extracted genes within specific pathways in NB and how they are involved in the inflammatory process.

We emphasize that this experimental work has demonstrated that neuroblastoma is not a "cold" tumor, and the originality lies precisely in this aspect. In fact, neuroblastoma manifests a significant inflammatory environment with the presence of genes that code for inflammatory cytokines (CCL2, CCL4, CCL5 , CCL8, CXCL10, etc.), for PTGS2 (responsible for the synthesis of PGE2) to which we must add genes responsible for the synthesis of collagens (COL6A2, COL6A3, COL12A1, COL14A1, COL5A1, COL5A2, etc.) that create with the microenvironment a cross-talk. Furthermore, it has been highlighted that the genes involved in Inflammation mediated by chemokine and cytokine signaling pathway and Integrin signaling pathway, describe not only an inflammatory condition but also describe an MES phenotype (PRRX1, NFATC2, RAC2, VAV1) of neuroblastoma.

  1. What does it add to the subject area compared with other published
    material? Authors aimed to demonstrate the inflammatory etiology of neuroblastoma.
  2. What specific improvements should the authors consider regarding the
    methodology? What further controls should be considered? Authors wrote along M%M section that they collect gene profiles of anonymous patients. Nevertheless, I think that significant information should be proposed to readers according to the neoplastic staging. Information about sex, age and other clinical information about patients are negligible but the tumor staging should be included since it can influence the results.
  3. Please describe how the conclusions are or are not consistent with the
    evidence and arguments presented. Please also indicate if all main questions
    posed were addressed and by which specific experiments. The conclusion section should be rephrased since, in the present form, it seems more the conclusive part of the Discussion section. 

Reply: We thank the reviewer for this suggestion. The sentences were reformulated, and clarification on data were provided. The scientific feedback that has emerged in the literature is certainly informative for the clinical implications that can be had, in fact the phenotypic state of MES, associated with a greater inflammatory characterisation and therefore with a possible immune vulnerability, can be the subject of clinical evaluation.

About the reviewer’s request on more detailed information on the patients enrolled, we would like to observe that the data obtained and present in the GSE16476 dataset are devoid of any reference to sex, age and stage, as these are untreated primary neuroblastoma patients whose profile was obtained from the residual material obtained during surgical procedures for diagnostic purposes. In any case, this anonymous condition provided a better assessment than that obtained without conditioning.

  1. Are the references appropriate? Yes.
    Reply: All references were checked and formatted according to the journal rule (…References may be in any style, provided that you use the consistent formatting throughout. It is essential to include author(s) name(s), journal or book title, article, or chapter title (where required), year of publication, volume and issue (where appropriate) and pagination. DOI numbers (Digital Object Identifier) are not mandatory but highly encouraged.)
  2. Please include any additional comments on the tables and figures and
    quality of the data.

Data, figures and graphs help the comprehension of the manuscript. 

Reply: All figures and tables were checked; some caption contents were enlarged to better clarify the contents of the graph and/or table.

L27- 30: rephrase: Neuroblastoma (NB) is an extracranial solid tumor, and it is considered the most frequent diagnosed neoplasia in children (7% 29 - 10%) after leukemia and brain tumors [1,2].

Reply: Done.

L30: do not start a sentence with an abbreviation (the same also for AA at L265)

Reply: Corrected.

L73: according to the sentence proposed by the authors, it seems that MHC-1 is the only marker to define a "cold tumor" but then, according to the Discussion section, it is not. Please, provide a good introduction in order to guide the readers to through the text properly. 

Reply: The sentence was reformulated, and clarifications were provided.

L90: the aim of the study is missing.

Reply: The sentence was reformulated, and clarifications were provided

L93: what authors mean with "untreated patients"? Have samples been collected prior surgery and chemo? Or patients never received any treatment? 

Reply: The sentences were reformulated, and clarification on data were provided.

L91-106: declare the tumor staging here if possible. 

Reply: The sentences were reformulated, and some explanations were provided.

L 106: rephrase: ... materials). All data were fully anonymized.

Reply: The sentences were reformulated, and a detailed explanations were provided. Particularly

L109: add a space prior (NMF)

Reply: Done.

L141: a sentence written in Italian is present. the same in L180-181

Reply: This has been fixed and solved.

L154: (r=2…,7) is it correct? 

Reply: This value is correct. The figure reports the plots of the consensus matrices obtained with different rank values (r=2…,7). From a deep analysis of the plots illustrating the Cophenetic Correlation Coefficients (CCC) and the consensus matrices, it was derived that the value k=5 is the first value at which the CCC trend be-gins to decrease. This allowed to choose k=5 as appropriate rank value for the considered data.

L213, 220, 222, 227, 250, 253: Axon and Inflammation must have a capital letter? 

Reply: Adjusted.

L239 and L 270: correct [23; 25] instead of [23], [25].

Reply: Done

L243: comma instead of full stop after B cell activation

Reply: Done

L348: This suggests that NB tumour cells instead of "This suggests that neuroblastoma (NB) tumour cells"

Reply: Done

L395: clinical implications? please, add some comments here. 

Round 2

Reviewer 1 Report

Comments and Suggestions for Authors

The paper was significantly improved following the first round of revision. 

Comments on the Quality of English Language

Although improved, there are still some non-native statements that impair clarity. However, these are now minor issues. 

Author Response

We thank the reviewer for having appreciate the revised manuscript. The English style has been checked by our language center to improve the readability of the paper and change some of the unclear statements. We hope this novel version is satisfactory.

Reviewer 2 Report

Comments and Suggestions for Authors

In the revised version of the manuscript, Boccarelli and colleagues made a number of important corrections that improved the quality of the article. It is very unfortunate that the authors ignored my suggestion to perform an independent verification of the data obtained by the authors using only computational biology methods. In my opinion, papers based only on a limited number of in silico algorithms give highly questionable results, which very often do not correspond to the results obtained later on in vivo and in patient material. To obtain more relevant results, such data should be double-checked by independent methods, not only by experimental but also by in silico approach. Dear authors, please prove the importance of the identified genes (PIK3CG, NFATC2, PIK3R2, VAV1, RAC2, COL6A2, COL6A3, COL12A1, COL14A1, ITGAL, ITGB7, FOS, PTGS2, PTPRC, ITPR3) for the pathogenesis of neuroblastoma by constructing and analyzing gene association networks. For this purpose, a gene association network based on the DisGeNET database (e.g. Childhood Neuroblastoma (CUI: C4086165) or Neuroblastoma (CUI: C0027819)) should be reconstructed using the STRING database (confidence score > 0.7) (or GeneMANIA) in Cytoscape, and then it should be determined whether the listed genes occupy hub positions (degree centrality, MCC or other characteristics). This approach will greatly strengthen your paper and give you additional methods to verify your data in subsequent articles. It would also greatly strengthen this paper if you could provide prognostic characteristics for the identified genes (Kaplan-Meier plots in neuroblastoma patients).

After these revisions, this paper can be considered for publication in IJMS.

Author Response

We would like to thank the reviewer for her profitable suggestion. We have added a novel section to the manuscript, namely “Gene association network analysis” which reports the independent verification of the data obtained by the computational biology methods used in the manuscript. To obtain additional results and checked what we have observed, we also reported the gene association networks provided by the GeneMANIA software (version 3.5.2), and the Search Tool for Retrieval of Interacting Genes (STRING) database (version 11.0). We mined the DisGeNET database using as disease strings both the Neuroblastoma (CUI: C0027819) and the Childhood Neuroblastoma (CUI: C4086165), and subsequently filtering the list of associated genes with the subset of genes PIK3CG, NFATC2, PIK3R2, VAV1, RAC2, COL6A2, COL6A3, COL12A1, COL14A1, ITGAL, ITGB7, FOS, PTGS2, PTPRC, ITPR3. In both cases (Neuroblastoma and Childhood Neuroblastoma diseases) the 8 genes PIK3CG, PTGS2, FOS, PTPRC, VAV1, ITPR3, NFATC2, PIK3R2. We have added two pictures Figure 5 and Figure 6 which depict the gene association network obtained by the cited software and related to these genes. As it could be observed from these figures, these evaluations have further strengthened the validity of these genes in the pathogenesis of Neuroblastoma, since these genes function as hubs, as well as are related to other genes whose roles is connected to the manifestation of the MES phenotype on an inflammatory basis.

Reviewer 3 Report

Comments and Suggestions for Authors

Dear Authors, Thank you for the extensive revision. I'm delighted about all the new explainations that you provided along the text. 

In my opinion, no further revisions are required. 

Author Response

We thank the reviewer for her kind judgments.

Round 3

Reviewer 2 Report

Comments and Suggestions for Authors

The authors have successfully responded to all my comments. With the use of gene networks, this article has become much more valid. The paper is ready to be published. I wish the authors continued success in their highly interesting research.